# ABHD17 proteins are novel protein depalmitoylases that regulate N-Ras palmitate turnover and subcellular localization

David Tse Shen Lin[1,2], Elizabeth Conibear[1,2]*

[1]Centre for Molecular Medicine and Therapeutics, University of British Columbia, Vancouver, Canada; [2]Department of Medical Genetics, University of British Columbia, Vancouver, Canada

**Abstract** Dynamic changes in protein S-palmitoylation are critical for regulating protein localization and signaling. Only two enzymes - the acyl-protein thioesterases APT1 and APT2 – are known to catalyze palmitate removal from cytosolic cysteine residues. It is unclear if these enzymes act constitutively on all palmitoylated proteins, or if additional depalmitoylases exist. Using a dual pulse-chase strategy comparing palmitate and protein half-lives, we found knockdown or inhibition of APT1 and APT2 blocked depalmitoylation of Huntingtin, but did not affect palmitate turnover on postsynaptic density protein 95 (PSD95) or N-Ras. We used activity profiling to identify novel serine hydrolase targets of the APT1/2 inhibitor Palmostatin B, and discovered that a family of uncharacterized ABHD17 proteins can accelerate palmitate turnover on PSD95 and N-Ras. ABHD17 catalytic activity is required for N-Ras depalmitoylation and re-localization to internal cellular membranes. Our findings indicate that the family of depalmitoylation enzymes may be substantially broader than previously believed.

*For correspondence: conibear@cmmt.ubc.ca

**Competing interests:** The authors declare that no competing interests exist.

## Introduction

Protein S-palmitoylation involves the post-translational attachment of the 16-carbon fatty acid palmitate to cysteine residues (*Conibear and Davis, 2010*; *Salaun et al., 2010*). While a survey of palmitoylation dynamics indicated the bulk of the palmitoyl-proteome is stably palmitoylated (*Martin et al., 2011*), rapid and constitutive palmitate turnover has been shown for several proteins, including the Ras GTPases, heterotrimeric G proteins, the neuronal post-synaptic density protein PSD95, and the Lck kinase (*Magee et al., 1987*; *Degtyarev et al., 1993*; *El-Husseini et al., 2002*; *Zhang et al., 2010*). Dynamic changes in palmitoylation modulate protein localization and trafficking and can be regulated in response to cellular signaling (*Conibear and Davis, 2010*).

Palmitoylation is mediated by a family of DHHC (Asp-His-His-Cys) proteins (*Greaves and Chamberlain, 2011a*), whereas the only enzymes identified to date that remove palmitate from cytosolic cysteines, the acyl-protein thioesterases (APTs) APT1 and APT2, are related members of the metabolic serine hydrolase (mSH) superfamily (*Duncan and Gilman, 1998*; *Tomatis et al., 2010*; *Long and Cravatt, 2011*). The β-lactone core-containing compound Palmostatin B (PalmB) potently inhibits these enzymes and blocks depalmitoylation of N-Ras and other proteins (*Dekker et al., 2010*; *Rusch et al., 2011*). Hexadecyl fluorophosphonate (HDFP) inhibits a subset of mSHs including APT1 and APT2 and also suppresses palmitate turnover (*Martin et al., 2011*). However, it is unclear if APT1 and APT2 are the only palmitoylthioesterases responsible for the depalmitoylation of cytosolic proteins (*Davda and Martin, 2014*).

**eLife digest** Proteins play important roles in many processes in cells. Some of these proteins can be modified by the addition of a molecule called palmitate. This process, termed "palmitoylation", helps direct these proteins to the compartments within the cell where they are needed to carry out their roles. One target of palmitoylation is N-Ras, which is a protein that can promote the development of cancer.

We understand quite a lot about how palmitate is added to proteins, but much less about how it is removed. So far, researchers have only identified two enzymes – known as APT1 and APT2 – that can remove palmitate from proteins, but it is possible that there are others. Identifying other "depalmitoylase" enzymes could help us find ways to block the removal of palmitate from N-Ras, which could lead to new treatments for some cancers.

Lin and Conibear used several biochemical techniques to search for depalmitoylase enzymes in human cells. The experiments reveal that although APT1 and APT2 are important for removing palmitate from some proteins, they are not needed to remove palmitate from N-Ras. Instead, Lin and Conibear found that an enzyme called ABHD17 removes palmitate from N-Ras. The next step following on from this work will be to find out what other proteins ABHD17 acts on in cells. A longer-term challenge will be to develop specific chemicals that inhibit ABHD17 activity and test if they are able to reduce the growth of cancer cells.

Here, we show that APT1 and APT2 inhibition or knockdown reduces palmitate turnover on some substrates but has no effect on N-Ras and PSD95. We identified members of the ABHD17 family as novel PalmB targets that depalmitoylate N-Ras and promote its relocalization to internal membranes. This demonstrates the enzymes responsible for protein depalmitoylation are more diverse than previously believed, which has important implications for understanding the selectivity and regulation of dynamic palmitate turnover.

## Results and discussion

APT1 and APT2 were proposed to act universally and constitutively to remove mislocalized proteins from intracellular membranes and allow their re-palmitoylation at the Golgi (*Rocks et al., 2010*). Reported rates of palmitate turnover on different substrates vary dramatically (*Qanbar and Bouvier, 2004*; *Martin et al., 2011*). We used a dual-click chemistry pulse-chase scheme to simultaneously measure palmitate and protein turnover of proteins expressed in COS-7 cells and labeled with the palmitate analogue 17-octadecynoic acid (17-ODYA) and the methionine surrogate L-azidohomoalanine (L-AHA) (*Martin and Cravatt, 2009*; *Zhang et al., 2010*). N-Ras had a rapid palmitate turnover as previously reported (*Figure 1A*; *Magee et al., 1987*). SNAP25 turned over slowly, whereas the glutamate decarboxylase subunit GAD65 and PSD95 had intermediate rates of depalmitoylation, demonstrating that these neuronal proteins undergo palmitate turnover at comparable rates in COS-7 cells or neuronal lines (*Greaves and Chamberlain, 2011b*; *El-Husseini et al., 2002*). A palmitoylated N-terminal fragment of Huntingtin (N-HTT) implicated in the pathogenesis of Huntington's disease (*Yanai et al., 2006*) also showed an intermediate palmitate turnover (*Figure 1B*). Treatment with the APT1/2 inhibitor PalmB inhibited the depalmitoylation of these substrates without affecting protein turnover (*Figure 1A,B*). In contrast, we found three proteins identified in a global palmitoyl-proteomics analysis (SPRED2, GOLIM4, and ITM2B) (*Martin et al., 2011*) did not undergo significant palmitate turnover, suggesting the apparent PalmB-resistant decline in palmitate labeling was due to protein instability (*Figure 1B*). These results confirm that proteins have inherently distinct rates of depalmitoylation, potentially reflecting differential recognition by APTs (*Lin and Conibear, 2015*). In all cases examined, PalmB inhibited the palmitate turnover of dynamically palmitoylated proteins.

APT1 and APT2 are reported to have differential substrate specificity (*Tomatis et al., 2010*; *Tian et al., 2012*). We found that the selective inhibitors C83 and C115, which target APT1 and APT2 respectively (*Adibekian et al., 2012*), had little effect on N-HTT depalmitoylation when used individually but achieved significant inhibition when applied together (*Figure 2A,B*). A similar effect was observed on GAD65 (*Figure 2—figure supplement 1A*). Surprisingly, these inhibitors had no

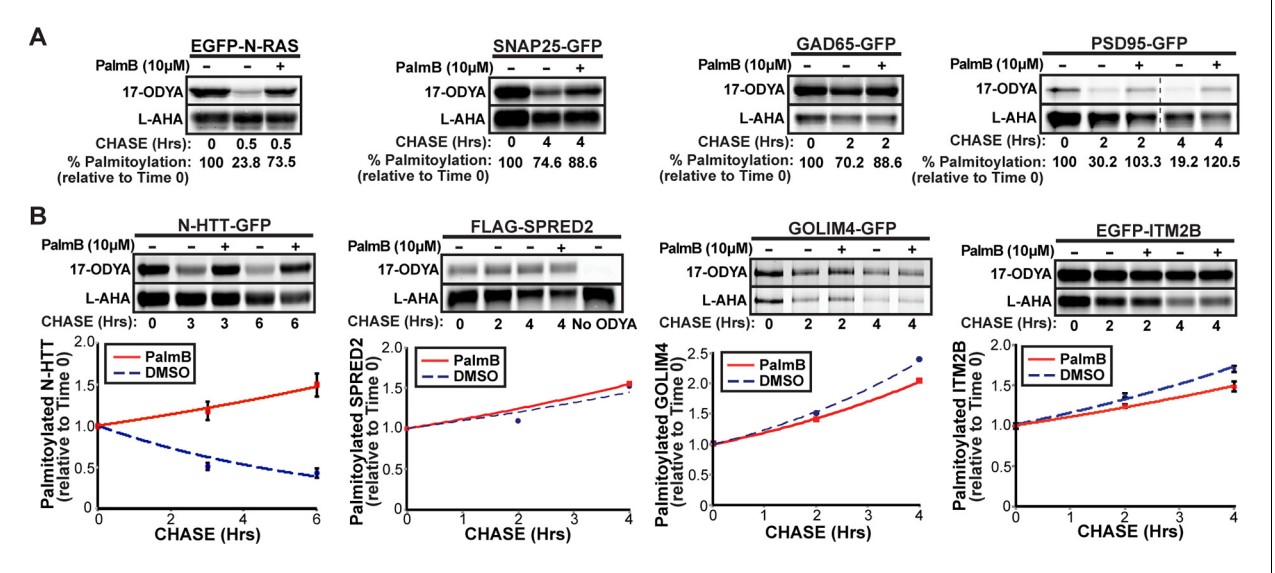

**Figure 1.** Dual-click chemistry labeling reveals differences in protein depalmitoylation dynamics. (**A**) Pulse-chase analysis of established palmitoyl-proteins (N-Ras, SNAP25, GAD65, PSD95) by dual-click chemistry in the presence of DMSO (-) or 10 μM PalmB (+). Representative in-gel fluorescence scans illustrate dual detection of 17-ODYA (palmitate analogue) and L-AHA (methionine analogue) using Alexa Fluor 488 and Alexa Fluor 647, respectively. Dashed line indicates cropping of a single gel. n = 2 per substrate. (**B**) Pulse-chase analysis of palmitate turnover on N-HTT, SPRED2, GOLIM4, and ITM2B by dual-click chemistry as described in (**A**). Upper panels: representative in-gel fluorescence scans; Lower panels: Time course of substrate depalmitoyation in DMSO- and PalmB-treated cells after normalizing 17-ODYA to L-AHA signals at each chase time. n = 2, mean ± SEM. 17-ODYA, 17-octadecynoic acid; L-AHA, L-azidohomoalanine; SEM, standard error of the mean.

effect on PSD95 or N-Ras depalmitoylation when used alone (*Figure 2—figure supplement 1B,C*) or together (*Figure 2C,D*). Double RNAi knockdown of APT1 and APT2 significantly inhibited N-HTT depalmitoylation (*Figure 2B*) and also reduced palmitate turnover on GAD65 (*Figure 2—figure supplement 1D*) but not PSD95 or N-Ras (*Figure 2C,D*). These findings, which are consistent with a recent report showing APT1/2-independent depalmitoylation of R7BP (*Jia et al., 2014*), strongly suggest that although APT1 and APT2 are responsible for depalmitoylating some proteins (N-HTT, GAD65), depalmitoylation of other cellular substrates, including PSD95 and N-Ras, involves other enzymes.

Previous studies suggested that APT1, APT2, and PPT1 were the sole mSHs targeted by PalmB (*Rusch et al., 2011*), whereas HDFP inhibited additional mSHs (*Martin et al., 2011*). In pulse-chase experiments, we found HDFP robustly inhibited the depalmitoylation of N-Ras, PSD95, and N-HTT (*Figure 3A-C*). Because palmitate removal from N-Ras and PSD95 does not require APT1 or APT2, their depalmitoylation may be mediated by a distinct mSH that is a common target of both PalmB and HDFP. To identify overlapping targets, we defined a set of 19 candidate mSHs that showed >25% inhibition by HDFP (*Supplementary file 1*; *Martin et al., 2011*) but excluded known proteases and mSHs with established luminal activity. We added to this list APT1L, which was previously implicated in BK channel depalmitoylation (*Tian et al., 2012*) but whose HDFP sensitivity was unknown. The PalmB sensitivity of each enzyme was evaluated by a competitive activity-based protein profiling (cABPP) assay, in which binding of an inhibitor occludes the enzyme active site and prevents labeling with the activity probe fluorophosphonate-rhodamine (FP-rho) (*Figure 3D*; *Kidd et al., 2001*). As expected, PalmB significantly reduced FP-rho labeling of both APT1 and APT2 (*Figure 3E,H*). In contrast, it had little effect on the labeling of seven candidates (*Figure 3F,H*), highlighting the distinct substrate specificities of PalmB and HDFP. Four mSHs did not label with FP-Rho due to low activity or expression and could not be assessed (*Supplementary file 1*). Notably, PalmB potently inhibited seven candidates: FASN, PNPLA6, ABHD6, ABHD16A, and ABHD17A/B/C (*Figure 3G,H*). Thus, PalmB has additional serine hydrolase targets beyond APT1 and APT2 that may function as protein depalmitoylases.

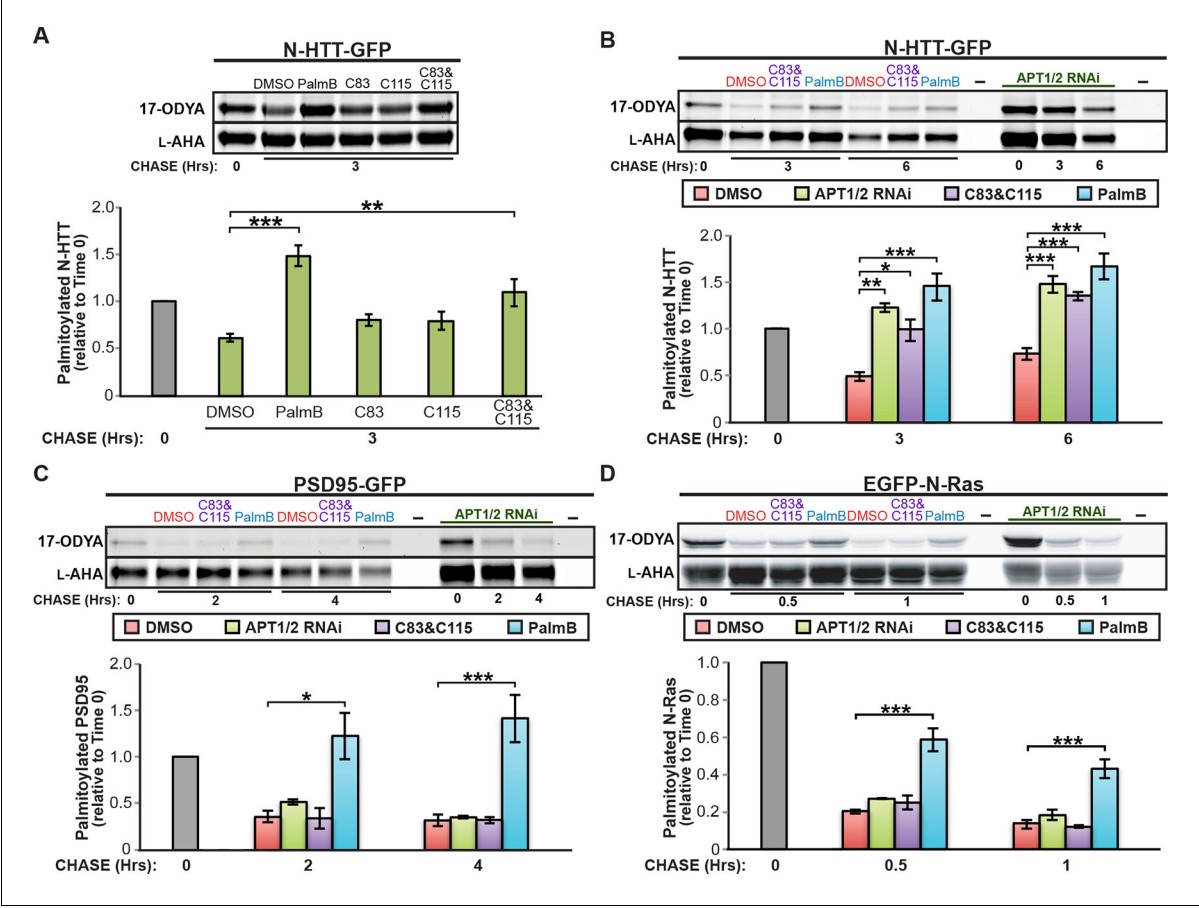

**Figure 2.** Downregulation of APT1 and APT2 inhibits HTT depalmitoylation but does not affect palmitate turnover on PSD95 or N-Ras. (**A**) Pulse-chase analysis of N-HTT palmitoylation in the presence of DMSO, 10 μM PalmB, 10 μM APT1-selective inhibitor C83, and/or 10 μM APT2-selective inhibitor C115, as described in *Figure 1*. n = 3, mean ± SEM. (**B-D**) Pulse-chase analysis of (**B**) N-HTT, (**C**) PSD95, and (**D**) N-Ras after APT1 and APT2 knockdown ("APT1/2 RNAi"), treatment with DMSO, treatment with 10 μM C83 and 10 μM C115, or treatment with 10 μM PalmB, as described in *Figure 1*. n = 3, mean ± SEM. *p < 0.05; **p < 0.01; ***p < 0.001. SEM, standard error of the mean.

The following figure supplement is available for figure 2:

**Figure supplement 1.** Downregulation of APT1 and APT2 inhibits GAD65 depalmitoylation but does not affect palmitate turnover on PSD95 or N-Ras.

The set of candidates inhibited by both PalmB and HDFP (*Figure 3G,H*) includes ABHD6, which associates with PSD95-containing complexes at synapses (*Schwenk et al., 2014*), and FASN, which functions in palmitoyl-CoA synthesis (*Wakil, 1989*). However, treatment with the ABHD6 inhibitor WWL70 (*Li et al., 2007*) or the FASN inhibitor C75 (*Kuhajda et al., 2000*) did not alter PSD95 depalmitoylation (*Figure 3—figure supplement 1A,C*). Palmitate turnover on PSD95 was also unaffected by RHC-80267, which moderately inhibited ABHD6 and PNPLA6 (*Figure 3—figure supplement 1B, D*; *Hoover et al., 2008*). Thus, ABHD6, PNPLA6, and FASN are unlikely to play a primary role in PSD95 depalmitoylation.

Selective inhibitors that target the remaining four candidates have not been identified. Therefore, we used pulse-chase click chemistry to test if increased expression of these enzymes enhances palmitate turnover. High levels of ABHD16A, ABHD6, or APT1/2 had little effect on N-Ras (*Figure 4A*) or PSD95 (*Figure 4—figure supplement 1A*) depalmitoylation. Strikingly, however, expression of ABHD17A, ABHD17B, or ABHD17C accelerated palmitate cycling on these proteins (*Figure 4A*, *Figure 4—figure supplement 1A*), strongly suggesting the uncharacterized ABHD17 family of mSHs are novel protein depalmitoylases.

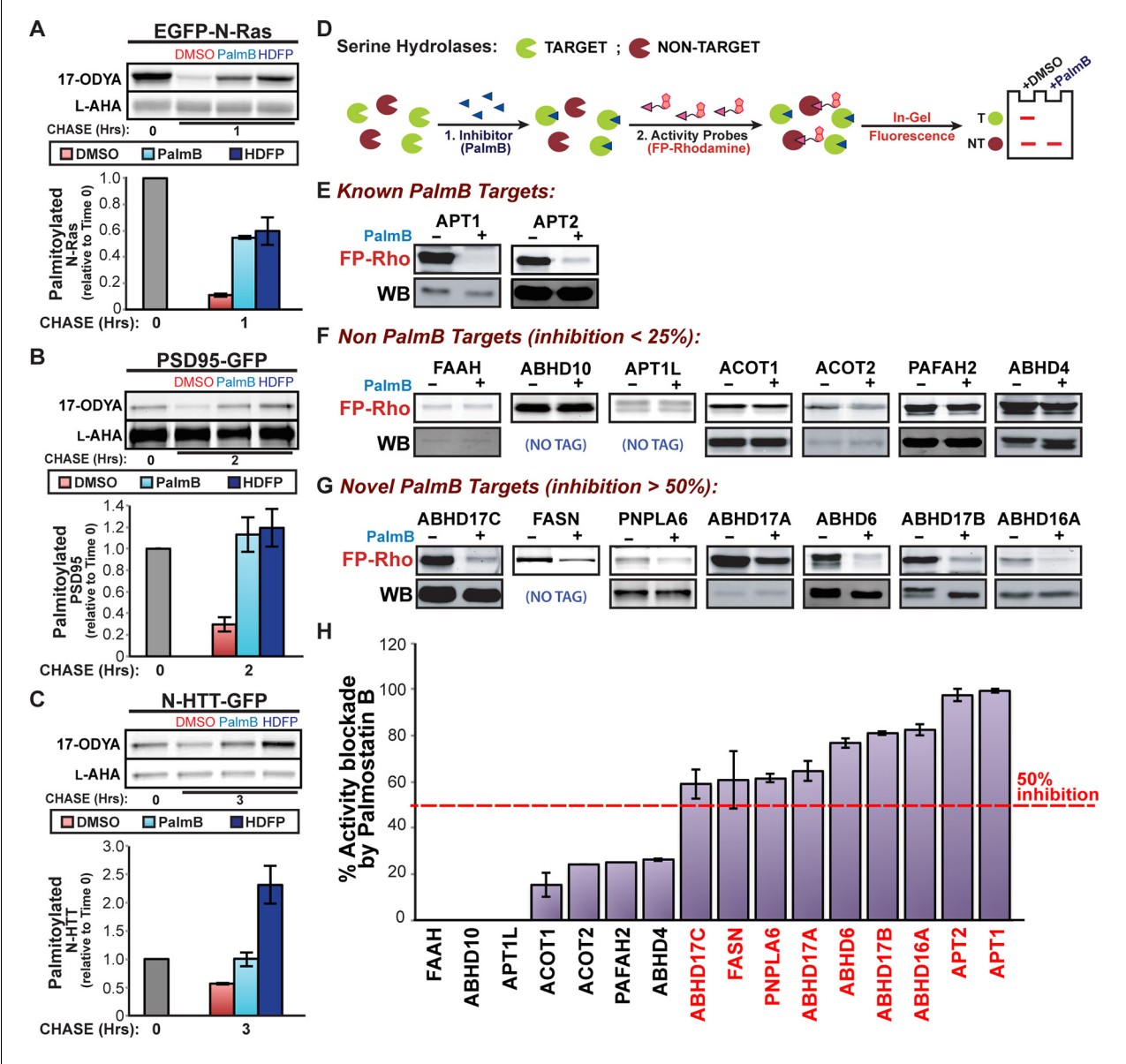

**Figure 3.** Shared targets of Palmostatin B and HDFP identified by competitive activity-based protein profiling. (**A-C**) Pulse-chase analysis of (**A**) N-Ras, (**B**) PSD95, and (**C**) N-HTT in the presence of DMSO, 10 μM PalmB or 20 μM lipase inhibitor HDFP as described in *Figure 1*. n = 3 (DMSO and PalmB) or 2 (HDFP), mean ± SEM. (**D**) Schematic diagram of the competitive ABPP assay used in this study. (**E-G**) Competitive ABPP of PalmB by in-gel fluorescence (FP-Rho). 16 HDFP targets were incubated with 2 μM FP-Rho in the presence (+) or absence (-) of 10 μM PalmB. Western blots (WB) show reduced FP-Rho labeling is not due to protein loss. (**H**) Percent inhibition of each HDFP target by PalmB. n = 3, mean ± SEM. Candidate depalmitoylases (>50% inhibition by PalmB) are highlighted in red. SEM, standard error of the mean,

The following figure supplement is available for figure 3:

**Figure supplement 1.** Treatment with serine hydrolase inhibitors WWL70, C75, and RHC-80267 does not affect PSD95 palmitate turnover.

We focused on ABHD17A, which showed the strongest effect in promoting palmitate turnover on N-Ras and PSD95. The ABHD17 proteins are targeted to membranes by a palmitoylated N-terminal cysteine cluster (*Kang et al., 2008*; *Martin and Cravatt, 2009*). We found ABHD17A localized to the plasma membrane and to Rab5- and Rab11-positive endosomes (*Figure 4—figure supplement 2A*). Mutation of the predicted active site serine (S211A) (*Figure 4B*) abolished ABHD17A activity (*Figure 4C*) but did not alter its localization (*Figure 4—figure supplement 2C*), whereas removing

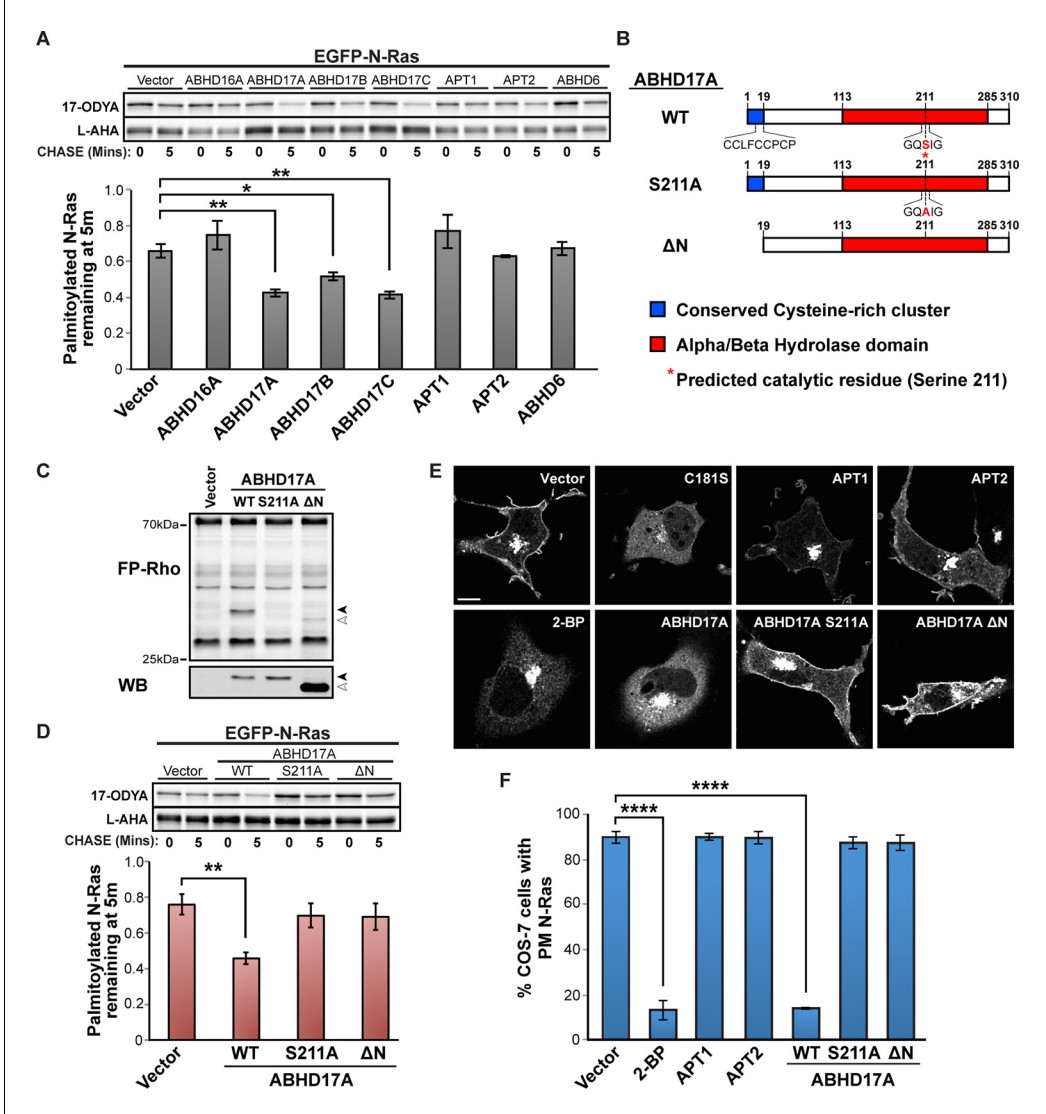

**Figure 4.** ABHD17A expression promotes N-Ras depalmitoylation and alters N-Ras subcellular localization. (**A**) Pulse-chase analysis of N-Ras co-expressed with candidate mSHs as described in *Figure 1*. n = 3, mean ± SEM. (**B**) Schematic of the ABHD17A wild type, catalytically-inactive (S211A), and N-terminal truncation (ΔN) mutant proteins used in this study. (**C**) ABPP of ABHD17A wild type and mutant proteins by in-gel fluorescence (FP-Rho). Western blot (WB) shows proteins expressed in each condition. Filled arrowheads: ABHD17A WT and S211A; Open arrowheads: ABHD17A ΔN. (**D**) Pulse-chase analysis of N-Ras co-expressed with ABHD17A wild type and mutant proteins as described in *Figure 1*. n = 3, mean ± SEM. (**E**) Representative live confocal images of EGFP-N-Ras-C181S and EGFP-N-Ras localization in COS-7 cells treated with 100 µM 2-bromopalmitate (2-BP) or co-expressing the indicated thioesterases. Scale Bar = 10 µm. (**F**) Bar graph representing percentage of COS-7 cells with plasma membrane EGFP-N-Ras under each condition studied in (**E**). n = 3 (100 cells counted per trial), mean ± SEM. *p < 0.05; **p < 0.01; ****p < 0.0001. mSHs, metabolic serine hydrolases; SEM, standard error of the mean.

The following figure supplements are available for figure 4:

**Figure supplement 1.** ABHD17 expression promotes PSD95 depalmitoylation.

**Figure supplement 2.** ABHD17A is localized to the plasma membrane and endosomal compartments.

the N-terminal amino acid residues 1-19 (ΔN; *Figure 4B*) shifted it to the cytosol (*Figure 4—figure supplement 2B,C*) and reduced its catalytic activity (*Figure 4C*). Importantly, neither mutant stimulated N-Ras or PSD95 depalmitoylation (*Figure 4D*, *Figure 4—figure supplement 1B*), suggesting both the catalytic activity and membrane localization of ABHD17A are functionally important.

We next examined the cellular consequences of ABHD17A expression. Disrupting N-Ras palmitoylation by mutating the palmitoylated residue (C181S) or treating cells with the inhibitor 2-bromopalmitate (2-BP) relocalized N-Ras from the plasma membrane to internal organelles, as previously described (*Choy et al., 1999*; *Goodwin et al., 2005*) (*Figure 4E,F*). Overexpression of APT1 or APT2 had little effect on N-Ras localization (*Figure 4E,F*), consistent with a recent report (*Agudo-Ibáñez et al., 2015*). In contrast, overexpression of ABHD17A, but not catalytically dead or cytosolic mutant forms, redistributed N-Ras from the plasma membrane to intracellular compartments consistent with its altered palmitoylation status (*Figure 4E,F*). Taken together, these findings demonstrate the membrane-localized pool of ABHD17A depalmitoylates N-Ras and alters its subcellular targeting.

To determine if the endogenous ABHD17 proteins regulate palmitate cycling in vivo, we investigated the effect of ABHD17 knockdown on N-Ras depalmitoylation in HEK293T cells. RT-qPCR (Reverse transcription quantitative polymerase chain reaction) showed efficient silencing of ABHD17A alone, or ABHD17A, ABHD17B, and ABHD17C in concert, after 72 hr with siRNA treatment (*Figure 5A*). ABHD17A knockdown had a slight effect on N-Ras depalmitoylation (p=0.084). In contrast, N-Ras palmitate turnover was significantly inhibited when all three ABHD17 proteins were simultaneously downregulated (p=0.0083), and this was not further enhanced by the APT1 and APT2 inhibitors C83 and C115 (*Figure 5B*). Knockdown was less effective than PalmB treatment, which could be due to activity of the residual ABHD17 enzymes. PalmB may also inhibit additional factors that either directly or indirectly affect N-Ras palmitate cycling. Taken together, these results demonstrate that ABHD17 proteins redundantly mediate palmitate turnover on N-Ras.

Our discovery that ABHD17 proteins are novel protein depalmitoylases expands the current repertoire of cellular APTs, and suggests depalmitoylation occurs in a substrate-selective and compartment-specific manner. Whereas APT1 and APT2 were proposed to act ubiquitously (*Rocks et al., 2010*; *Vartak et al., 2014*), ABHD17-mediated depalmitoylation of N-Ras at the plasma membrane may specifically attenuate oncogenic signaling pathways (*Song et al., 2013*). ABHD17 proteins are

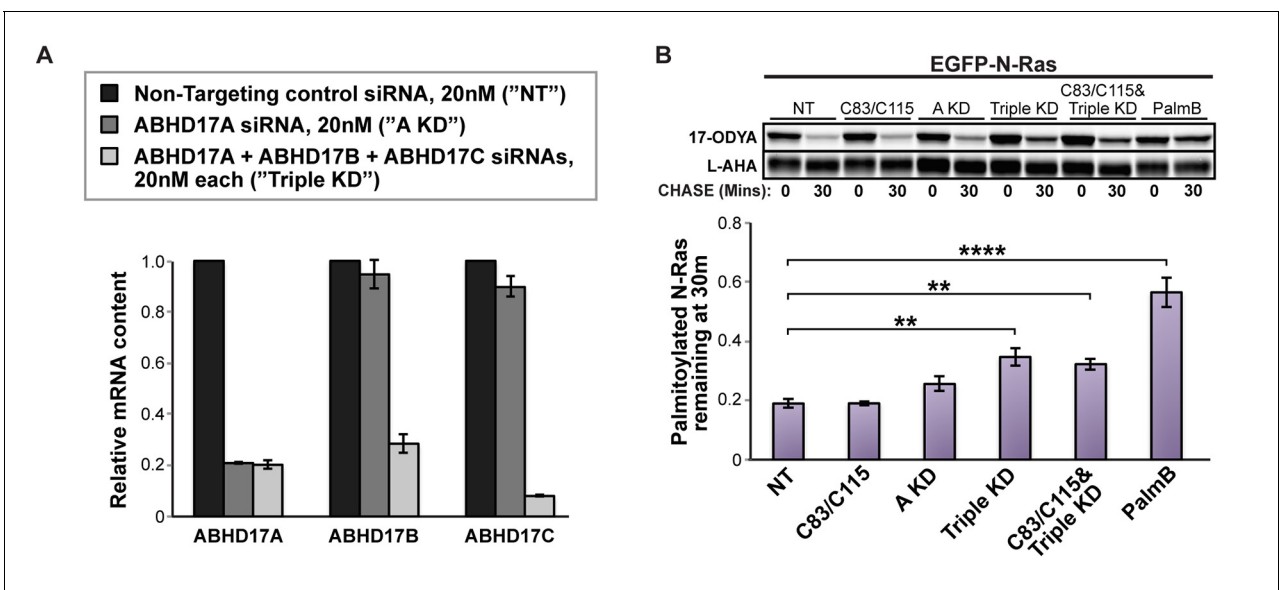

**Figure 5.** Simultaneous knockdown of ABHD17 isoforms inhibits N-Ras palmitate turnover. (**A**) RT-qPCR of ABHD17A, ABHD17B, and ABHD17C transcript levels in HEK 293T cells treated with Non-Targeting siRNA ("NT", black), ABHD17A siRNA alone ("A KD", gray), or ABHD17A/ ABHD17B/ ABHD17C siRNAs ("Triple KD", light gray) for 72 hr. n = 3, mean ± SEM. (**B**) Pulse-chase analysis of N-Ras palmitoylation in siRNA-transfected HEK 293T cells treated with vehicle (DMSO), 10 µM C83 and C115, or 10 µM PalmB as described in *Figure 1*. n = 3, mean ± SEM. **p < 0.01; ****p < 0.0001. SEM, standard error of the mean.

also active in the brain (*Bachovchin et al., 2010*), where palmitoylated PSD95 regulates AMPA (α-amino-3-hydroxy-5-methyl-4-isoxazolepropionic acid) receptor nanodomain assemblies linked to synaptic plasticity (*Fukata et al., 2013*). miRNA-138 targets APT1 to alter dendritic spine size (*Siegel et al., 2009*), whereas the *Caenorhabditis elegans* ABHD17 homologue AHO-3 regulates starvation-induced thermotactic plasticity (*Nishio et al., 2012*). Thus, functionally specialized APTs may prove to be critical modulators of palmitoyl-proteins in distinct cellular processes.

The total number of cellular depalmitoylases is not known. We identified new PalmB targets, consistent with a recent report showing PalmB inhibits ABHD12 and monoacylglyerol lipase (*Savinainen et al., 2014*). As the mSH superfamily consists of >110 members, only half of which are functionally annotated (*Simon and Cravatt, 2010*), a comprehensively survey the mSH proteome may uncover yet more depalmitoylases. APTs are a critical element of the dynamic palmitoylation cycle, thus it will be imperative to identify the complete set of cellular APTs and determine how they contribute to the regulation of dynamic palmitoylation.

# Materials and methods

## Plasmids and siRNAs

Plasmids expressing EGFP-N-Ras, PSD95-GFP, N-HTT-GFP, SNAP25-GFP were provided by Dr. Michael Hayden (University of British Columbia). Plasmids expressing Myc-hAPT1, GOLIM4-GFP, FLAG-SPRED2, and GAD65-GFP were generous gifts from Dr. Takashi Izumi (Gunma University), Dr. Adam Linstedt (Carnegie Mellon University), Dr. Akihiko Yoshimura (Keio University), and the late Dr. Alaa El-Husseini (University of British Columbia), respectively. Venus-tagged Rab5, Rab7, and Rab11 plasmids were gifts from Dr. Nevin Lambert (Georgia Regents University). EGFP-ITM2B was cloned by polymerase chain reaction (PCR) amplification of the ITM2B ORF (open reading frame) from MGC Fully Sequenced Human BRI3 cDNA, clone ID 3163436 (OpenBiosystems; Mississauga, ON), using the forward primer 5′-ATTTAACCCGGGGATGGTGAAGATTAGCTTCCAGCC-3′ and the reverse primer 5′-ATTTAAGGTACCTCACACCACCCCGCAGAT-3′, followed by restriction digest and ligation with BspEI/KpnI-digested pEGFP-C3 vector from Clontech (Mountain View, CA). EGFP-N-Ras-C181S was generated by Quikchange mutagenesis (Stratagene; La Jolla, CA) using the forward primer 5′-CAACAGCAGTGATGATGGTACCCAGGGTAGTATGGGATTGCCATGTGTGG-3′ and the reverse primer 5′-CCACACATGGCAATCCCATACTACCCTGGG TACCATCATCACTGCTGTTG-3′ with EGFP-N-Ras as the template.

For cloning of mSHs for activity-profiling studies, plasmids containing corresponding human ORFs were purchased from DNASU (Arizona State University, Tempe, AZ) and OpenBiosystems, or obtained as clones from the hORFeome v8.1 Collection (*Yang et al., 2011*). Genes of interest were amplified by PCR using oligos with flanking restriction sites (described in *Supplementary file 2*), and the resulting mSH-encoding PCR products were subcloned into vectors of interest (FLAG-NT, generously provided by Dr. Stefan Taubert, University of British Columbia; or pCINeo, Promega [Madison, WI]).

The ABHD17A-FLAG construct was used as the template to generate ABHD17A mutant and mCherry-tagged plasmids. S211A-FLAG in pCINeo was generated by Quikchange mutagenesis, and ABHD17A ΔN-FLAG was amplified by PCR then subcloned into pCINeo. ABHD17A-mCherry wild type and mutant plasmids were generated by pairing each forward oligo with the reverse ABHD17A-mCherry-Linker oligo as listed in *Supplementary file 2*. The resulting ABHD17A fragments were fused with the PCR-amplified C-terminal mCherry cassette by overlapping extension PCR (OEPCR) and subcloned into pCINeo vector with EcoRI and XbaI. Similarly, mCherry-APT1 and mCherry-APT2 plasmids were constructed by fusing the N-terminal mCherry cassette with PCR-amplified APT1 and APT2 fragments using OEPCR and subcloning the resulting fragments into pCINeo vector with EcoRI and XbaI.

The pSUPER vector and the shRNA pSUPER-APT1 plasmid used in knockdown studies was a generous gift from Dr. Gerhard Schratt (University of Marburg), and ON-TARGET*plus* SMARTpool siRNAs targeting APT2, ABHD17A, ABHD17B, or ABHD17C, as well as Non-Targeting control siRNA, were purchased from Dharmacon (Lafayette, CO).

## Chemicals

Lipofectamine 2000, Lipofectamine RNAiMax, sodium dedocyl sulfate (SDS) solution, L-azidohomoalanine (L-AHA), Alexa Fluor 488-azide (AF488-az), Alexa Fluor 647-alkyne (AF647-alk), TRIzol reagent, and Prolong Gold Antifade Mountant with DAPI were purchased from Life Technologies (Burlington, ON). X-tremeGENE 9 was purchased from Roche (Indianapolis, IN). Palmostatin B was purchased from Merck Scientific (Billerica, MA). Tris[(1-benzyl-1$H$-1,2,3-triazol-4-yl)methyl]amine (TBTA), Tris(2-carboxyethyl)phosphine hydrochloride (TCEP), Triton-X 100 (TX-100), sodium deoxycholate, $CuSO_4$, palmitic acid, and 2-bromopalmitate were obtained from Sigma-Aldrich (St. Louis, MO). 17-ODYA, C75, WWL70, and RHC-80267 were purchased from Cayman Chemical (Ann Arbor, MI). HDFP, C83, and C115 were gifts from Dr. Brent Martin (University of Michigan), and FP-rhodamine was generously provided by Dr. Benjamin Cravatt (Scripps Institute).

## Cell culture conditions

COS-7 and HEK293T/17 cells from ATCC (Manassas, VA) were maintained and propagated in high-glucose Dulbecco's Modified Eagle Medium (DMEM) supplemented with 10% fetal bovine serum (FBS; Life Technologies), 4 mM L-glutamine and 1 mM sodium pyruvate, in a humidified incubator at 37°C, 5% $CO_2$.

## cDNA and siRNA transfections

For pulse-chase metabolic studies and activity-based protein-profiling studies, COS-7 cells were transfected with cDNAs as indicated in each experiment using Lipofectamine 2000 as per manufacturer's instructions. Cells were grown in six-well plates (Corning; Corning, NY) and transfected at 90% confluence with 1 μg of cDNA per well for pulse-chase analyses with inhibitors, or 2 μg cDNA per well for pulse-chase analyses with thioesterase overexpression. For immunofluorescence studies, COS-7 cells were grown on glass coverslips (Fisher; Pittsburg, PA) in 24-well plates (Corning) and transfected at 60–90% confluence with 0.5 μg of cDNA per well using Xtreme-GENE 9 according to product instructions. Experiments involving small molecules were carried out 20–24 hr following transfection, and experiments involving co-expression of candidate mSHs were carried out 24–48 hr post-transfection, as described below.

For APT1 and APT2 studies, a double knockdown approach was used (*Bond et al., 2011*) where COS-7 cells were transfected with siRNA (100 nM final concentration per transfection) on days 1 and 3 with 5 μL of Lipofectamine 2000 per transfection. One microgram of cDNA was co-transfected with the siRNA on day 3, and pulse-chase studies were carried out on day 4, 20 hr following the co-transfection. For ABHD17 studies, HEK293T cells were transfected on day 1 with siRNA in 9 μL Lipofectamine RNAiMax, and on day 3 with 1μg of EGFP-N-Ras in 4 μL Lipofectamine 2000. Pulse-chase and RT-qPCR studies were performed on day 4, 20 hr following cDNA transfection.

## Pulse-chase metabolic labeling with inhibitors

Twenty hours following transfection, COS-7 cells or HEK293T cells were washed twice in phosphate-buffered saline (PBS) and starved in cysteine- and methionine-free DMEM containing 5% charcoal-filtered FBS (Life Technologies) for 1 hr. Cells were then labeled with 30 μM 17-ODYA and 50 μM L-AHA for 1.5 hr in this media. The labeling media was removed, and cells were briefly washed twice in PBS before chasing in complete DMEM supplemented with 10% FBS and 300 μM palmitic acid. Small molecule inhibitors or DMSO (vehicle) were added at chase time 0. At indicated time points, cells were washed twice in PBS and lysed with 500 μL triethanolamine (TEA) lysis buffer (1% TX-100, 150 mM NaCl, 50 mM TEA pH 7.4, 2×EDTA-free Halt Protease Inhibitor [Life Technologies]). The lysates were transferred to 1.5 mL Eppendorf tubes (Corning), vigorously shaken (3 × 20s) while placed on ice in between each agitation. Lysates were cleared by centrifugation at 16,000× g for 15 min at 4°C. Solubilized proteins in the supernatant were quantified using Bicinchoninic acid (BCA) assay (Life Technologies) and subsequently used for immunoprecipitations as described below.

## Immunoprecipitations

For immunoprecipitations, Protein A or Protein G sepharose beads (GE Healthcare; Mississauga, ON) were washed thrice in TEA lysis buffer. Protein A beads were pre-incubated with rabbit anti-GFP antibodies (Life Technologies) and Protein G beads were pre-incubated with FLAG M2

antibodies (Sigma-Aldrich) for 2 hr at 4°C, before the addition 500 µg – 1 mg of transfected COS-7 cell lysates containing indicated proteins. Immunoprecipitations were carried out for 12–16 hr on an end-to-end rotator at 4°C. Following immunoprecipitation, sepharose beads were washed thrice in modified RIPA buffer (150 mM NaCl, 1% sodium deoxycholate (w/v), 1% TX-100, 0.1% SDS, 50 mM TEA pH7.4) before proceeding to sequential on-bead CuAAC/click chemistry.

## Sequential on-bead CuAAC/click chemistry

Sequential on-bead click chemistry of immunoprecipitated 17-ODYA/L-AHA-labeled proteins was carried out as previously described (Zhang et al., 2010), with minor modifications. After immunoprecipitation, sepharose beads were washed thrice in RIPA buffer, and on-bead conjugation of AF488 to 17-ODYA was carried out for 1 hr at room temperature in 50 µL of freshly mixed click chemistry reaction mixture containing 1 mM TCEP, 1 mM $CuSO_4 \cdot 5H_2O$, 100 µM TBTA, and 100 µM AF488-az in PBS. After three washes in 500 µL RIPA buffer, conjugation of AF647 to L-AHA was carried out for 1 hr at room temperature in 50 µL click-chemistry reaction mixture containing 1 mM TCEP, 1 mM $CuSO_4 \cdot 5H_2O$, 100 µM TBTA, and 100 µM AF647-alk in RIPA buffer. Beads were washed thrice with RIPA buffer and resuspended in 10 µL SDS buffer (150 mM NaCl, 4% SDS, 50 mM TEA pH7.4), 4.35 µL 4× SDS-sample buffer (8% SDS, 4% Bromophenol Blue, 200 mM Tris-HCl pH 6.8, 40% Glycerol), and 0.65 µL 2-mercaptoethanol. Samples were heated for 5 min at 95°C, and separated on 10% tris-glycine SDS-PAGE gels for subsequent in-gel fluorescence analyses.

## Competitive activity-based protein profiling

Twenty-four hours following transfection with mSH constructs, COS-7 cells were washed twice in PBS, transferred to a new vial by scraping in PBS, and lysed by gentle sonication on ice. Protein was quantified by BCA assay. Thirty micrograms of total protein was incubated either with DMSO or small molecule inhibitors at indicated concentrations at room temperature for 30 min, prior to the addition of FP-Rho (2 µM final concentration). Labeling reactions were carried out at room temperature for 1 hr and quenched with 4× SDS-sample buffer heated to 95°C for 5 min. Samples were separated on SDS–PAGE, analyzed by in-gel fluorescence, then transferred onto nitrocellulose membrane for Western blotting.

## In-gel fluorescence analyses

A Typhoon Trio scanner (GE Healthcare) was used to measure in-gel fluorescence of SDS–PAGE gels: AF488 signals were acquired using the blue laser (excitation 488 nm) with a 520BP40 emission filter, AF647 signals were acquired using the red laser (excitation 633 nm) with a 670BP30 emission filter, and rhodamine signals were acquired with the green laser (excitation 532 nm), with a 580BP30 emission filter. Signals were acquired in the linear range and quantified using the ImageQuant TL7.0 software (GE Healthcare). For pulse-chase analyses, the ratio of palmitoylated substrates were calculated as AF488/AF647 values at each time point, normalized to the value at T=0.

## Western blotting

Nitrocellulose membranes were blocked with PBS with 0.1% Tween-20 (PBST) containing 3% bovine serum albumin (BSA, Sigma) for 1 hr, and incubated with primary antibodies (rabbit anti-GFP, 1:1,000; or mouse anti-FLAG M2, 1:1,000) in PBST + 3% BSA for 2 hr, followed by 3x15 min washes with PBST + 0.3% BSA. Membranes were then incubated with secondary antibodies (IRDye 800CW goat anti-mouse IgG,1:10,000; or IRDye 680RD goat anti-rabbit IgG,1:10,000) (LI-COR Biosciences; Lincoln, NE) in PBST + 0.3% BSA for 1 hr. After three washes in PBST, membranes were imaged using the LI-COR Odyssey Scanner (LI-COR). Signals were acquired in the linear range using the 680nm and 800nm lasers and quantified using the Image Studio software (LI-COR).

## Confocal microscopy and EGFP-N-Ras localization

COS-7 cells were co-transfected with EGFP-N-Ras and empty vector or indicated mCherry-tagged thioesterases at a 1:1 ratio (total 0.5 µg DNA per well) in Lab-Tek 8-well chamber slides (Fisher). Twenty-four hours post-transfection, cells were imaged on a TCS SP8 confocal laser scanning microscope (Leica Microsystems; Mannheim, Germany), and EGFP-N-Ras localization was quantified by counting 100 cells per experiment.

## Immunocytochemistry

Twenty hours post-transfection, cells were washed twice with PBS, and fixed in 4% paraformaldehyde (PFA) solution (4% PFA, 4% sucrose in PBS) for 20 min. Cells were permeabilized for 1 min in PBS containing 0.1% TX-100, washed thrice in PBS, and blocked with PBS +3% BSA for 60 min before incubating with primary antibodies (mouse anti-FLAG-M2, 1:500; rabbit anti-FLAG (Sigma), 1:200; or mouse anti-GM130 (BD Biosciences; San Jose, CA), 1:200) for 1 hr. Coverslips were washed thrice and incubated with secondary antibodies (goat anti-mouse Alexa Flour 488 and goat anti-rabbit Alexa Fluor 594 (Life Technologies), 1:1000 each) for an hour. Coverslips were washed with PBS and mounted on glass slides with ProLong Gold Antifade Mountant containing DAPI. Cells were observed with an Axioplan 2 fluorescence microscope (Carl Zeiss; Oberkochen, Germany) using a Plan-Apochromat 100× 1.40 NA oil immersion objective lens. Images were acquired with a Cool-SNAP camera (Roper Scientific; Planegg, Germany) using YFP, GFP, and Texas Red filters and Meta-Morph 7.7 software (MDS analytical Technologies; Toronto, ON), and adjusted using Metamorph 7.7.

## RNA extraction, reverse transcription, and RT-qPCR

Seventy-two hours post-transfection with siRNA pool(s), HEK293T cells were collected in 1 mL TRIzol reagent. Samples were snap-frozen at -80°C until used. Total RNA extraction was carried out with PureLink RNA Mini kit (Life Technologies) following manufacturer instructions. For each sample, 1 μg of RNA was used to synthesize cDNA with QuantiTect Reverse Transcription Kit (Qiagen; Hilden, Germany). RT-qPCR was performed in 15 μL reactions using a Rotor-Gene 6000 (Qiagen) and PerfeCTa SYBR Green FastMix (Quanta Biosciences; Gaithersburg, MD) with gene-specific primer pairs listed in *Supplementary file 3*. ABHD17 mRNA levels were determined by the $\Delta\Delta$Ct method normalizing to β-actin mRNA levels. PCR efficiencies of primers were examined by standard curve of serial-diluted untreated whole cell samples.

## Statistical analyses

Statistical analyses were carried out by performing Student's two-tailed t-tests using Prism 6 (Graph-Pad Software, Inc., La Jolla, CA), with DMSO-treated (*Figure 2* and *Figure 3*), vector-co-transfected (*Figure 4*), or Non-targeting siRNA-transfected (*Figure 5*) samples as the control group. All significant differences (p< 0.05) are indicated in the figures.

# Acknowledgements

We thank Dr. Nicholas Davis for critical comments; Drs. Takashi Izumi, Gerhard Schratt, Adam Linstedt, Akihiko Yoshimura, Nevin Lambert, Brent Martin and Benjamin Cravatt for reagents; and Phoebe Lu and Nikita Verheyden for assistance with RT-qPCR.

# Additional information

### Funding

| Funder | Grant reference number | Author |
|---|---|---|
| Canadian Institutes of Health Research | Institute of Genetics Team Grant, GPG-202165 | Elizabeth Conibear |
| Canada Foundation for Innovation | Leading Edge Fund, 30636 | Elizabeth Conibear |
| University of British Columbia | Four Year Fellowship (FYF) | David Tse Shen Lin |
| Canadian Institutes of Health Research | New Investigator Salary Award | Elizabeth Conibear |

The funders had no role in study design, data collection and interpretation, or the decision to submit the work for publication.

## Author contributions
DTSL, Conception and design, Acquisition of data, Analysis and interpretation of data, Drafting or revising the article; EC, Conception and design, Analysis and interpretation of data, Drafting or revising the article

## Author ORCIDs
David Tse Shen Lin, http://orcid.org/0000-0001-5695-9446

# Additional files

### Supplementary files
• Supplementary File 1. List of Metabolic serine hydrolases inhibited by HDFP. A summary table compiling the 29 serine hydrolases targeted by HDFP (>25% activity inhibition) as determined by cABPP-SILAC (Stable isotope labeling of amino acids in culture) in (*Martin et al., 2011*). LYPLAL1 (APT1L) was added to this list as a candidate enzyme for Palmostatin B testing (*Tian et al., 2012*).

• Supplementary File 2. List of cloning oligos used in this study. A table listing PCR primers used to subclone candidate serine hydrolases for cABPP, pulse-chase/click chemistry, and confocal imaging studies.

• Supplementary File 3. List of gene-specific RT-qPCR primer pairs used in this study. A table listing gene-specific primer pairs for verification of transcript levels in HEK293T cells by RT-qPCR in *Figure 5A*.

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
