## [Decision Letter]

Thank you for submitting your work entitled "ABHD17A is a novel protein depalmitoylase that regulates N-Ras palmitate turnover and subcellular localization" for consideration by *eLife*. Your article has been reviewed by three peer reviewers, one of whom is a member of our Board of Reviewing Editors, and Tony Hunter as the Senior Editor.

The reviewers have discussed the reviews with one another and the Reviewing editor has drafted this decision to help you prepare a revised submission.

Summary:

The manuscript by Lin and Conibear investigates the flux of Cys palmitoylation in various proteins. Using a combination of tagging reagents for palmitoylation (17-ODYA) and protein levels (L-AHA) in pulse chase experiments, the authors investigate various inhibitors of known cytosolic depalmitoylases. They find somewhat surprisingly that Ras, previously believed to be depalmitoylated by APT1 and APT2, is not actually affected by these enzymes. This was shown using both knockdown of these enzymes by RNAi (with nice positive controls) as well as more selective inhibitors of these enzymes than the previously reported PalmB. The evidence directly challenges the prior conclusions that APT1 is the target of palmostatin B mediated inhibition of H/Nras depalmitoylation (Dekker, et al. Nat Chem Biol. 2010 Jun;6(6):449-56), which relied on less robust strategies of acyl-biotin exchange chemistry and indirect surrogates of palmitoylation using fluorescence protein subcellular localization. The authors then search for the real Ras depalmitoylase, using chemoproteomic and overexpression strategies and make a reasonably strong case that ABHD17A is a Ras (and PSD95) depalmitoylase. This is demonstrated by overexpression of wt but not Ser mutant or N-terminal deletion mutant (which mislocalizes) ABHD17A. Overexpression of ABHD17A, ABHD17B, ABHD17C and ABHD6 suggested the first 3 serine hydrolyses could decrease palmitoylation levels of PSD-95. Additional site-directed mutagenesis studies demonstrated that the catalytic residue and plasma membrane localization of ABHD17A was required for depalmitoylation of PSD-95-GFP and EGFP-N-Ras in COS-7 cells. ABHD17A localization and active-site mutation also affected the cellular distribution of EGFP-N-Ras in COS-7 cells. This is interesting work with important implications for the palmitoylation field. The clarity this new evidence provides is essential to guide efforts that will define and potentially interrupt the contribution of protein palmitoylation to Nras driven malignancies including melanoma and leukemias. It is by and large technically strong and innovative, nicely taking advantage of a series of elegant chemical tools most of which were previously reported by Martin and Cravatt. The manuscript in a general sense seems appropriate for *eLife*, as a short paper, however I do have several concerns that should be considered.

Essential revisions:

1) A missing experiment is an RNAi knockdown (or genetic knockout) of ABHD17A which could/should induce palmitoylation of Ras and PSD95 according to the authors' model. This would be especially helpful since no selective inhibitor appears to be available for this enzyme. If the authors have already done this experiment, they should comment on the results. If they have not, they should discuss why not.

2) Interestingly, an analysis of Hras, which is dually palmitoylated, is conspicuously absent in the present study and this is notable largely because of the direct challenge made toward the prior conclusions regarding APT1 activity on both H- and Nras. The authors should make an effort to address their choice to avoid including an analysis of Hras.

Minor points:

1) Although not essential, it would be of some value to assess the depalmitoylation of Ras palmitoylated peptides by ABHD17A as well as APT1 and APT2 to see if the cellular results can be observed in a purified system.

2) I am not really sure what is meant by the dashed lines and the cropping of lanes in several of the blots. Given that some of the effects are modest (ca. less than 2-fold), it would be better to just repeat experiments and represent on one complete gel.

3) Figure 3 ABHD4 looks like it may be tagged by PalmB when taking loading into account.

4) In Figure 4 panels D and E, it appears that the baseline levels of palmitoylation of Ras and PSD95 are elevated in S211A and DeltaN. Can the authors comment on whether this is true and if so why it might be occurring?

---

## [Author Response]

Essential revisions:

*1) A missing experiment is an RNAi knockdown (or genetic knockout) of ABHD17A which could/should induce palmitoylation of Ras and PSD95 according to the authors' model. This would be especially helpful since no selective inhibitor appears to be available for this enzyme. If the authors have already done this experiment, they should comment on the results. If they have not, they should discuss why not.*

Our previous results indicated each of the ABHD17 proteins can depalmitoylate N-Ras, suggesting simultaneous knockdown of all three isoforms might be required (or quintuple knockdown, if they function redundantly with APT1/2). We had not previously attempted this experiment due to its inherent challenges, but in response to the reviewers’ request, we have now investigated palmitate turnover on N-Ras after single knockdown of ABHD17A, or triple knockdown of ABHD17A/B/C in the presence or absence of APT1/2-selective inhibitors. We present these results in a new Figure 5, and discuss them in a new paragraph in Results and Discussion, paragraph 8. Importantly, we observed significant inhibition of N-Ras depalmitoylation after knockdown of all three ABHD17 isoforms (Figure 5), demonstrating the ABHD17 proteins are critical mediators of N-Ras depalmitoylation in vivo.

*2) Interestingly, an analysis of Hras, which is dually palmitoylated, is conspicuously absent in the present study and this is notable largely because of the direct challenge made toward the prior conclusions regarding APT1 activity on both H- and Nras. The authors should make an effort to address their choice to avoid including an analysis of Hras.*

We chose to focus our analysis on the singly-palmitoylated N-Ras, because the two palmitate modifications on H-Ras differentially affect H-Ras trafficking and localization (Misaki et al., J. Cell Biol., 2010), and could be regulated by distinct APTs. To address the reviewers’ comments, we have now conducted a preliminary analysis of wild type H-Ras palmitoylation indicating that overexpressed ABHD17A can target H-Ras (see Figure 6). Dissecting the relative roles of ABHD17 and APT1 in regulating turnover at each palmitoylation site, and the corresponding effects on H-Ras membrane localization and lipid microdomain association, will require additional long-term studies.10.7554/eLife.11306.015Author Response Image 1.**DOI:**
http://dx.doi.org/10.7554/eLife.11306.015

We do not believe our results necessarily contradict previous findings. Many prior studies used Palmostatin B as an indirect means to examine APT1 activity on H- and N-Ras in cultured cells, and based on our results, it is likely many effects attributed to APT1 were due to ABHD17. Only a few studies have directly measured Ras palmitoylation after APT1 knockdown or overexpression. Dekker et al. (Nat. Chem. Biol., 2010) reported APT1 knockdown increased steady state palmitoylation of N-Ras by acyl biotin exchange assay (p=0.15), whereas Agudo-Ibáñez et al. (Mol. Cell Biol., 2015) showed overexpression of APT1 significantly reduced steady state [3H]-palmitate labeling of H-Ras (p<0.005). In contrast, we specifically measured depalmitoylation rates using a pulse-chase protocol that detects rapid palmitate turnover. Our results demonstrate that ABHD17 isoforms play a more important role than APT1/2 in regulating dynamic palmitate cycling on specific substrates including N-Ras, but do not preclude a function for APT1/2 in regulating steady-state palmitoylation of H-Ras.

Minor points:

*1) Although not essential, it would be of some value to assess the depalmitoylation of Ras palmitoylated peptides by ABHD17A as well as APT1 and APT2 to see if the cellular results can be observed in a purified system.*

We agree an *in vitro* palmitoylation assay would be valuable. However, this is not a trivial experiment because we have shown ABHD17A must be membrane-anchored to display full activity (Figure 4). A more thorough analysis of how membrane association affects ABHD17A activity will be required before pursuing this approach.

*2) I am not really sure what is meant by the dashed lines and the cropping of lanes in several of the blots. Given that some of the effects are modest (ca. less than 2-fold), it would be better to just repeat experiments and represent on one complete gel.*

The images in the original figures represented single gels that had been cropped to remove extraneous lanes. All quantitation was performed on the uncropped gels, thus we believe cropping had no effect on the results. Nevertheless, we repeated the experiments in Figure 3 as suggested, and our revised Figure 3 shows the new, complete gels. For Figure 3—figure supplement 1 and Figure 1 (PSD95 panel) we would be happy to provide uncropped gels on request.

*3) Figure 3 ABHD4 looks like it may be tagged by PalmB when taking loading into account.*

Some binding to ABHD4 is possible, as our analysis showed PalmB reduced the activity of ABHD4 by approximately 25%, consistent with our observation that PalmB binds additional mSHs. However, to find the most significant PalmB targets, we chose the set of enzymes showing >50% inhibition for more detailed study.

*4) In Figure 4 panels D and E, it appears that the baseline levels of palmitoylation of Ras and PSD95 are elevated in S211A and DeltaN. Can the authors comment on whether this is true and if so why it might be occurring?*

We clearly see no effect on the rate of Ras or PSD95 de-palmitoylation in the presence of these mutants, yet baseline palmitoylation does appear to be elevated in an activity-independent manner. This would be worth future investigation, but is beyond the scope of the current study.